# Population genomics of *Plasmodium ovale* species in sub-Saharan Africa

Kelly Carey-Ewend [1] ✉, Zachary R. Popkin-Hall [2], Alfred Simkin[3], Meredith Muller[2], Chris Hennelly[2], Wenqiao He[2], Kara A. Moser[2], Claudia Gaither[2], Karamoko Niaré[3], Farhang Aghakanian[2], Sindew Feleke [4], Bokretsion G. Brhane[4], Fernandine Phanzu[5], Melchior Mwandagalirwa Kashamuka[6], Ozkan Aydemir [7], Colin J. Sutherland [8], Deus S. Ishengoma[9,10], Innocent M. Ali [11], Billy Ngasala[12], Albert Kalonji[5], Antoinette Tshefu[6], Jonathan B. Parr [2,13,14], Jeffrey A. Bailey [3], Jonathan J. Juliano [1,2,13,14,16] & Jessica T. Lin[2,13,15,16]

*Plasmodium ovale curtisi* (*Poc*) and *Plasmodium ovale wallikeri* (*Pow*) are relapsing malaria parasites endemic to Africa and Asia that were previously thought to represent a single species. Amid increasing detection of ovale malaria in sub-Saharan Africa, we present a population genomic study of both species across the continent. We conducted whole-genome sequencing of 25 isolates from Central and East Africa and analyzed them alongside 20 previously published African genomes. Isolates are predominantly monoclonal (43/45), with their genetic similarity aligning with geography. *Pow* shows lower average nucleotide diversity ($1.8\times10^{-4}$) across the genome compared to *Poc* ($3.0\times10^{-4}$) ($p < 0.0001$). Signatures of selective sweeps involving the dihydrofolate reductase gene have been found in both species, as are signs of balancing selection at the merozoite surface protein 1 gene. Differences in the nucleotide diversity of *Poc* and *Pow* may reflect unique demographic history, even as similar selective forces facilitate their resilience to malaria control interventions.

Parasites in the genus *Plasmodium* were responsible for an estimated 249 million cases of malaria and 608,000 deaths in 2022[1]. Ninety-four percent of these cases occurred in the World Health Organization Africa Region, where control efforts have primarily focused on the predominant species, *P. falciparum* (*Pf*)[2]. Yet these case counts likely underrepresent the burden of non-falciparum species, which may be rising in prevalence even where control efforts have successfully reduced *P. falciparum* transmission[3–5]. Over the last few decades,

[1]Department of Epidemiology, Gillings School of Global Public Health, University of North Carolina, Chapel Hill, NC, USA. [2]Institute for Global Health and Infectious Diseases, University of North Carolina, Chapel Hill, NC, USA. [3]Department of Pathology and Laboratory Medicine, Brown University, Providence, RI, USA. [4]Ethiopian Public Health Institute, Addis Ababa, Ethiopia. [5]SANRU Asbl, Kinshasa, Democratic Republic of the Congo. [6]Kinshasa School of Public Health, Kinshasa, Democratic Republic of the Congo. [7]Program in Molecular Medicine, Chan Medical School, University of Massachusetts, Worcester, MA, USA. [8]London School of Hygiene and Tropical Medicine, London, UK. [9]National Institute for Medical Research (NIMR), Dar es Salaam, Tanzania. [10]Department of Biochemistry, Kampala International University in Tanzania, Dar es Salaam, Tanzania. [11]Department of Biochemistry, Faculty of Science, University of Dschang, Dschang, Cameroon. [12]Muhimbili University of Health and Allied Sciences, Dar es Salaam, Tanzania. [13]Division of Infectious Diseases, University of North Carolina School of Medicine, University of North Carolina, Chapel Hill, NC, USA. [14]Curriculum in Genetics and Molecular Biology, University of North Carolina School of Medicine, University of North Carolina, Chapel Hill, NC, USA. [15]Department of Microbiology and Immunology, University of North Carolina School of Medicine, University of North Carolina, Chapel Hill, NC, USA. [16]These authors jointly supervised this work: Jonathan J. Juliano, Jessica T. Lin. ✉e-mail: kelly_carey-ewend@med.unc.edu

genomic studies of *P. falciparum* have enabled monitoring of drug resistance markers[6], facilitated the identification of promising vaccine candidates[7], uncovered the structure of parasite populations[8], and identified evolutionary forces shaping their demography[9,10]. Much less is known about non-falciparum species, especially their comparative evolutionary history and susceptibility to malaria control interventions focused on *P. falciparum*.

*Plasmodium ovale* was first identified as a separate malaria species in 1922 based on the appearance of oval-shaped erythrocytes that contained non-ring parasite forms[11]. Hallmarks of this parasite species are its restriction to younger red cells and, therefore, the propensity to cause low-density infections, as well as relapses from liver hypnozoites, similar to *P. vivax* and *P. cynomologi*. The species often causes coinfection alongside *P. falciparum* which, along with its low parasite densities, makes it challenging to differentiate morphologically on peripheral blood smears[12]. The advent of polymerase chain reaction (PCR)-based diagnostics has improved the detection of ovale infections, but initial PCR surveys across Africa and Asia based on the small subunit rRNA gene revealed two apparent groups of *P. ovale* parasites, termed classic and variant[13]. The discovery of perfect sequence segregation of six genomic markers, and more recently, 12 mitochondrial loci, between classic and variant *P. ovale* isolates collected across Africa and Asia has led to the conclusion that this dimorphism actually represents a true species divide in the *P. ovale* clade[14,15]. The nomenclature of these species is currently evolving but will be referred to as *P. ovale curtisi* (*Poc*, formerly classic) and *P. ovale wallikeri* (*Pow*, formerly variant) herein[16–18].

*P. ovale curtisi* and *P. ovale wallikeri* have since been confirmed to circulate within the same human populations throughout Africa and Asia[19,20], with both detected by PCR at higher rates than previously appreciated[5,12,21,22]. Limited investigation of the genetic diversity and population genetics of the two *P. ovale* species have hinted at low diversity and/or small effective population size, as few unique haplotypes have been identified at antigenic gene targets like apical membrane antigen 1 (*ama1*) and merozoite surface protein 1 (*msp1*)[20,23]. There is some indication that drugs used to treat *P. falciparum* are also shaping *P. ovale* parasite populations; signs of a selective sweep involving a mutant *dhfr* allele (implicated in pyrimethamine resistance) have been detected in both *Poc* and *Pow*[24,25]. Until now, the low density of most *P. ovale* isolates combined with the lack of an in vitro culture system has hindered whole-genome sequencing of these parasites[11]. However, with the development of strategies for parasite DNA enrichment, as well as the construction of the first reference genomes in 2017, genome-wide analyses are now possible[15,26–28].

In this work, we employ hybrid capture or leukodepletion to enrich *P. ovale* spp. DNA and perform whole-genome sequencing (WGS) of 25 clinical isolates collected from studies conducted across Ethiopia, the Democratic Republic of the Congo, Tanzania, and Cameroon. Combined with 20 additional public whole genomes from 11 countries spanning East, Central, and West Africa, we seek to better understand the comparative biology of *P. ovale curtisi*, *P. ovale wallikeri*, and co-endemic *P. falciparum* by examining their complexity of infection, population structure, nucleotide diversity, and genomic signatures of selection.

## Results

### High-quality genomic coverage of African *P. ovale* isolates

Parasite samples from 25 *P. ovale*-infected individuals collected at ten sites spanning Ethiopia, the Democratic Republic of the Congo (DRC), Tanzania, and Cameroon were selected for whole-genome sequencing (Table 1)[29–34]. These included 13 *P. ovale curtisi* and 12 *P. ovale wallikeri* isolates that were selected from six studies based on robust amplification of the *po18S* rRNA gene (Ct < 36) and

**Table 1 | Studies of origin for 45 *P. ovale* isolates**

| Study | Country of origin | Year of collection | Study population | # selected for sequencing (# of *Poc* and *Pow*) |
|---|---|---|---|---|
| *hrp2/3* Deletion Survey[29] | Ethiopia | 2017–2018 | Febrile patients presenting to health facilities in the Amhara, Tigray, and Gambella regions | 7 (2 *Poc*, 5 *Pow*) |
| SANRU Rural Health Program[30] | Democratic Republic of the Congo | 2017 | Febrile patients presenting to health facilities in Sud-Kivu, Bas-Uele, and Kinshasa Provinces | 6 (2 *Poc*, 4 *Pow*) |
| Kinshasa Malaria Longitudinal Study[31] | Democratic Republic of the Congo | 2015–2017 | Members of households participating in longitudinal study of malaria | 5 (5 *Poc*) |
| TranSMIT[32] | Tanzania (East) | 2018–2022 | Asymptomatic children and adults attending school or health clinics in rural Bagamoyo district, eastern Tanzania | 4 (3 *Poc*, 1 *Pow*) |
| MSMT[21][33] | Tanzania (West) | 2020–2022 | Tanzanian citizens at health facilities | 2 (2 *Pow*) |
| Dschang Febrile Cohort[34] | Cameroon | 2020–2021 | Febrile patients presenting to health facilities in western Cameroon | 1 (1 *Poc*) |
| Joste et al., *JID* 2023[36] * | Cameroon, Senegal, Ivory Coast | 2013–2021 | *P. ovale* infections identified in France after travel to an endemic country | 4 (4 *Pow*) |
| Rutledge et al., *Nature* 2017[28] * | Ghana, Cameroon | Unknown | Symptomatic *P. falciparum* infected individuals with strong *P. ovale* signals among whole-genome sequencing results | 2 (1 *Poc*, 1 *Pow*) |
| Higgins et al., *Sci Rep.* 2024[15] * | Tanzania, Kenya, South Sudan, Congo, Cameroon, Nigeria, Sierra Leone | 2019–2020 | *P. ovale* infections identified in the United Kingdom after travel to an endemic country. | 11 (6 *Poc*, 5 *Pow*) |
| Ansari et al., *Int J Parasitol.* 2016[37] * | Gabon, Nigeria | Unknown | Febrile Chinese males presenting to clinics in Jiangsu Province, China, after travel to an endemic country | 3 (1 *Poc*, 2 *Pow*) |

*Raw sequencing data for these 20 isolates were directly incorporated into the analysis pipeline after retrieval from the European Nucleotide Archive or NCBI Sequence Read Archive.

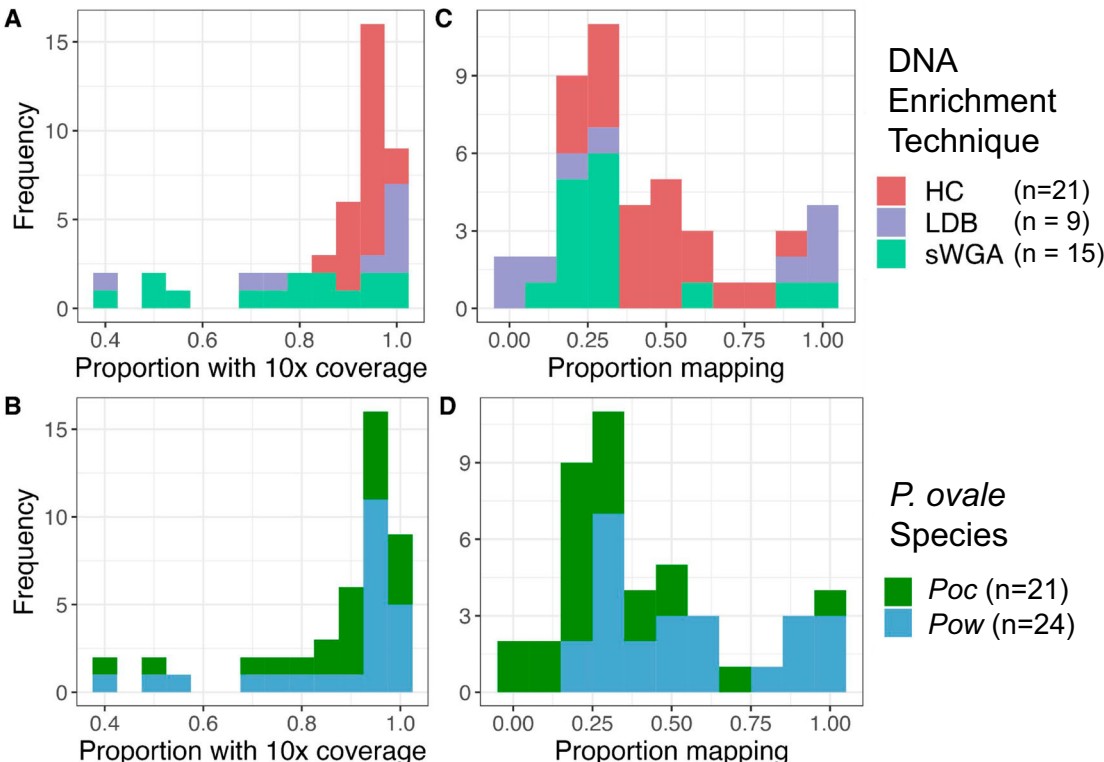

**Fig. 1 | Coverage and mapping among 45 isolates.** The proportion of the corresponding *P. ovale* reference genome covered by ≥10 reads by DNA enrichment technique (**A**) and species (**B**). The proportion of reads mapped to that reference genome by DNA enrichment technique (**C**) and species (**D**). For *Poc*, chromosome 10 was excluded due to incomplete coverage by hybrid capture baits. Five samples from Higgins et al. were not incorporated into this study due to having <30% 10x coverage of the corresponding genome. HC hybrid capture, LDB leukodepleted blood, sWGA selective whole-genome amplification. Source data are provided as a Source Data file.

predominance of one *ovale* species within each isolate. The majority of the samples ($n = 21$) underwent a custom-designed hybrid capture with RNA baits to preferentially isolate ovale DNA extracted from dried blood spots for sequencing, while four additional whole blood samples were leukodepleted (LDB) at the time of collection by CF11 filtration and directly sequenced without enrichment[35]. Finally, genomic data of 20 *P. ovale* isolates sequenced as part of four previously published studies were retrieved from the European Nucleotide Archive and the Sequence Read Archive[15,28,36,37]. These isolates either underwent selective whole-genome amplification (sWGA) or leukodepletion for parasite DNA enrichment prior to sequencing. Further data on all parasite isolates are found in Supplemental Table 1.

Whole-genome sequencing achieved high genome coverage, with an overall average of 86 and 87% tenfold coverage across the core genome for the 21 *P. ovale curtisi* and 24 *P. ovale wallikeri* isolates, respectively (Fig. 1). Coverage and mapping proportion were highest when aligned to the *P. ovale* reference genome determined by the *Poc/Pow* species-specific qPCR assay, corroborating initial species assignment. Compared to sWGA and LDB samples, the hybrid capture method used to enrich parasite DNA in the majority of samples yielded more complete coverage across all chromosomes except for chromosome 10 (Fig. 1). The hybrid capture was originally designed for *P. ovale wallikeri*, with additional *P. ovale curtisi* baits then selected to cover areas that differ between the two ovale genome assemblies (PowCR01 and PocGH01)[28]. Due to *Pow* chromosome 10 being incomplete in the PowCR01 reference genome (only 470 kb), this approach did not provide coverage for the full *Poc* chromosome 10 (1300 kb). This led to substantially lower coverage for chromosome 10 across all *Poc* hybrid capture isolates (40–60%

10x coverage vs. >85% for all other *Poc* chromosomes); thus, chromosome 10 was excluded from all genome-wide analysis in *Poc* isolates to limit error.

As expected, hybrid capture led to preferential sequencing of *P. ovale* DNA among samples that were co-infected with *P. falciparum* (*Pf*); *Pf*-positive isolates that underwent hybrid capture yielded only 2–11% 10x coverage of the *Pf* genome compared to >90% 10x coverage of the *Pf* genome among leukodepleted blood samples. For genomic analysis, insertions/deletions, multiallelic sites, low-quality variants, and variants within tandem repeats and expanded gene families were excluded (see Methods), yielding final biallelic single nucleotide polymorphism (SNP) call sets of 73,015 SNPs for *P. ovale curtisi* and 45,669 for *P. ovale wallikeri*.

## Low complexity of infection

Complexity of infection (COI), or the number of unique parasite clones present in a given isolate, was estimated 1000 times using THERE-ALMcCOIL for all 21 *P. ovale curtisi* and 24 *ovale wallikeri* isolates, as well as 2077 geographically matched *P. falciparum* isolates downloaded from the publicly available MalariaGEN Pf6 dataset (Fig. 2)[38]. Twenty out of 21 *Poc* isolates (95%) and 23 out of 24 *Pow* isolates (96%) were estimated to be monoclonal; the remaining isolate in each *P. ovale* species was found to comprise two parasite clones. By comparison, roughly half (1165/2077; 56%) of *P. falciparum* samples were monoclonal. COI differed significantly ($p = 0.005$) among the three *Plasmodium* species. In pairwise comparisons, both *Poc and Pow* had significantly lower COI compared to *P. falciparum* ($p = 0.001$ and $p = 0.004$, respectively).

The two multiclonal *P. ovale* infections were both hybrid capture-enriched, high coverage (94 and 98% tenfold coverage in *Poc* and *Pow*,

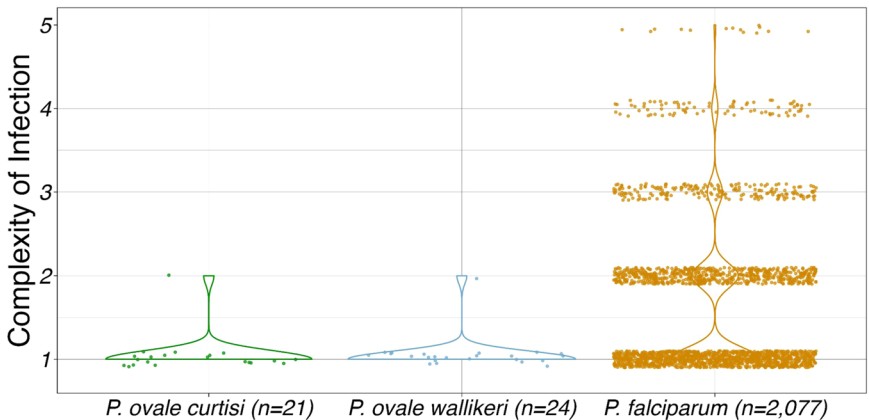

**Fig. 2 | Complexity of infection by *Plasmodium* species.** Median estimated complexity of infection (COI) among 21 *Poc* isolates, 24 *Pow* isolates, and 2077 *P. falciparum* isolates geographically matched to the *P. ovale* samples by country of origin. The distributions of COI differed significantly among the three species ($p < 0.0001$) by a Kruskal–Wallis test, with *Pow* and *Poc* showing significantly lower COI than *P. falciparum* ($p = 0.0004$ and $0.012$, respectively) in Dunn's multiple comparisons tests. The average read depth of coverage for *Poc*, *Pow*, and *Pf* isolates were 44.1, 95.7, and 147.5, respectively. Source data are provided as a Source Data file.

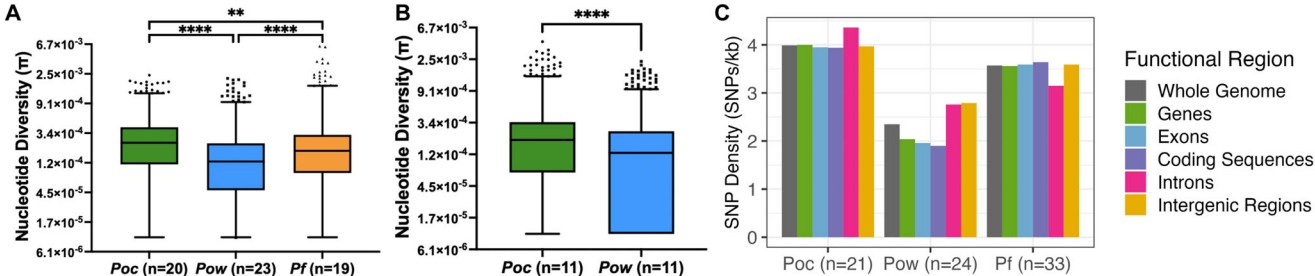

**Fig. 3 | Nucleotide diversity (π) of orthologous genes and SNP density by functional genomic region among *Poc*, *Pow*, and *P. falciparum* (*Pf*) isolates.** **A** Nucleotide diversity (π) per gene among 2008 sets of orthologous genes in monoclonal *Poc*, *Pow*, and *Pf* samples. Boxes denote the 25th, median, and 75th percentiles; whiskers are drawn at the 1st and 99th percentiles. π of 0 was coded as $1 \times 10^{-5}$ to plot on a logarithmic scale. Nucleotide diversity was significantly different between orthologues of all three species by two-sided Tukey's multiple comparisons tests, with *Poc* orthologues showing higher diversity than orthologues of *Pow* and *P. falciparum*, and *Pow* orthologues also showing lower diversity than those in *P. falciparum* (*p* values <0.001, =0.002, and <0.0001, respectively).

**B** Nucleotide diversity (π) per gene among 2911 sets of orthologous genes in geographically matched monoclonal *Poc* and *Pow* samples. Boxes denote the 25th, median, and 75th percentiles; whiskers are drawn at the 1st and 99th percentiles. π of 0 was coded as $1 \times 10^{-5}$ to plot on a logarithmic scale. Nucleotide diversity was significantly lower among *Pow* orthologues compared to *Poc* using a two-sided Wilcoxon's matched-pair signed rank test ($p < 0.0001$). **C** SNP density in different functional regions of the genome among all *Pow*, *Poc*, and *P. falciparum* isolates. SNP single nucleotide polymorphism, kb kilobase. Source data are provided as a Source Data file.

respectively) and came from high-transmission areas of the DRC[39]; each had a COI of 2. In order to determine whether the two clones in these samples were distinct lineages or meiotic siblings, we analyzed the distribution of heterozygous SNPs across the genome. We hypothesized that meiotic siblings would only have heterozygous SNPs in specific regions, reflecting recombination within the mosquito midgut[40]. In both samples, after filtering to high-confidence SNPs based on population-wide allele frequency, we saw an even distribution of heterozygous SNPs across the genome, suggestive of two distinct parasite lineages in the same host rather than meiotic siblings (Supplemental Fig. 1).

**Lower nucleotide diversity in *P. ovale wallikeri* compared to *P. ovale curtisi***

Among a collection of 3339 sets of one-to-one orthologous genes between the *Poc*, *Pow*, and *P. falciparum* genomes, we identified 2008 sets that achieved high-quality sequencing coverage and had no overlap with masked genomic regions in any of the three species. The average species-specific nucleotide diversity (π) among these orthologues in the 20 monoclonal *Poc*, 23 monoclonal *Pow*, and 19 geographically matched monoclonal *Pf* samples were $2.9 \times 10^{-4}$, $1.8 \times 10^{-4}$,

and $2.6 \times 10^{-4}$, respectively. These were significantly different between species ($p < 0.0001$, $F = 98$, df = 2), with orthologues in *P. ovale curtisi* more diverse than in *P. ovale wallikeri* and *P. falciparum* (*p* values <0.0001 and 0.002, respectively), and *Pow* orthologues less diverse than in *Pf* ($p < 0.0001$) (Fig. 3A). To mitigate bias by geographic coverage and orthology with the *P. falciparum* genome, we repeated this analysis using 2,911 *Poc-Pow* orthologues among a group of geographically matched monoclonal *Poc* and *Pow* samples ($n = 11$ each, Supplemental Table 3), revealing average nucleotide diversities of $2.5 \times 10^{-4}$ and $1.8 \times 10^{-4}$, respectively (Fig. 3B). Nucleotide diversity was still significantly lower in *Pow* orthologues compared to *Poc* ($p < 0.0001$).

This high nucleotide diversity in *P. ovale curtisi* was consistent with an investigation of the total number and density of genome-wide SNPs. Variant calling and filtering resulted in almost twice the number of SNPs for *P. ovale curtisi* (73,015) as for *P. ovale wallikeri* (45,669), despite a slightly smaller number of *Poc* isolates (21 vs. 24). This corresponded to a higher density of SNPs across the *Poc* genome (4.0 SNPs per kilobase[kb] in *Poc* vs. 2.4 SNPs/kb in *Pow*). However, a smaller proportion of SNPs in the *Poc* genome were nonsynonymous mutations, as the ratio of nonsynonymous-to-synonymous (dN/dS) mutations was 1.5 and 2.5 in *Poc* and *Pow*,

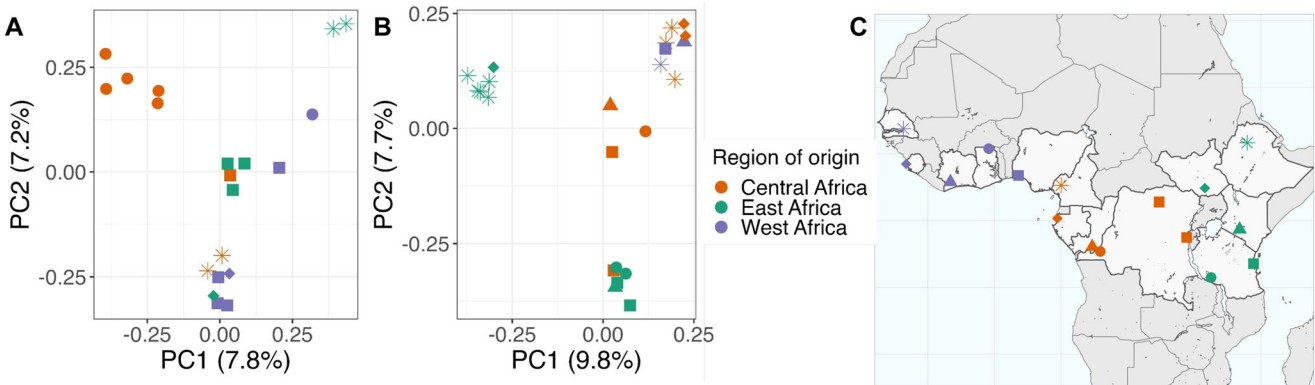

**Fig. 4 | Principal component analysis of monoclonal *P. ovale* spp. isolates.** Principal component analysis showing the first two principal components among 20 monoclonal *Poc* isolates (**A**) and 23 monoclonal *Pow* isolates (**B**) using 4116 and 3189 biallelic SNPs, respectively. Samples are colored by region of country of origin; in the map, parasites from travelers are assigned to the capital city (**C**). In PC2 of *Pow*, SNPs within both the *ts-dhfr* and *mrp1* gene were among the top 0.5% of contributors. Source data are provided as a Source Data file.

respectively. Among the aforementioned geographically matched *P. ovale* isolates, dN/dS within 2911 orthologous genes was 1.4 for *Poc* and 2.5 for *Pow*, consistent with the broader estimates. SNP densities in both the *Poc* and *Pow* genomes were lowest in protein-coding sequences (3.9 and 1.9 SNPs/kb, respectively), and higher in introns (4.4 and 2.8 SNPs/kb) (Fig. 3C). Intergenic regions in *Poc* showed relatively lower SNP density similar to protein-coding sequences (4.0 SNPs/kb), but these same regions in *Pow* had relatively high SNP density similar to introns (2.8 SNPs/kb).

## Parasite genomic similarity recapitulates geographic relationships

Genome-wide principal component (PC) analysis of the monoclonal samples of each ovale species revealed the spatial arrangement of related parasites along PC1 and PC2 that aligns with their location of origin (Fig. 4). These components accounted for 15.0 and 17.5% of the genetic differentiation in *Poc* and *Pow*, respectively. While cluster analysis by ADMIXTURE found the best fit when modeling each isolate as a separate cluster, except for one pair of isolates per species originating from the capital of Kinshasa in the DRC (in *Poc*) and the Amhara region in Ethiopia (in *Pow*), geographic alignment was evident in the PCA. For *Pow*, PC1 and PC2 appear to reflect an East-West axis and North-South axis, respectively, with samples from Ethiopia and South Sudan in the west divided from others by PC1. In the PCA for *Poc*, Ethiopian parasites were also organized separately from other samples, as did isolates from Kinshasa in the DRC. The remaining *Poc* samples show some division between East, Central, and West Africa, though the alignment with geography is less consistent than in *Pow*. In both *P. ovale* spp., PC3 and PC4 further separated samples from various countries (Supplemental Fig. 2).

Examination of the top 0.5% of variants by contribution to each of the first 4 principal components revealed that SNPs within genes encoding multidrug resistance protein 1 (*mdr1*) and dihydrofolate reductase-thymidylate synthase (*dhfr-ts*), two putative antimalarial resistance genes, were major contributors to the North-South axis in *Pow* PC2. Among all 24 *Pow* samples, three previously documented haplotypes in *Pow dhfr-ts*, a key gene in folate metabolism that is implicated in pyrimethamine resistance[41,42], appear to drive this geographic differentiation, with the Phe57Leu + Ser58Arg haplotype existing in 45% of our Central African clones and 36% of our East African clones but none of the sequenced West African clones (Supplemental Fig. 3A)[25]. This haplotype is associated with resistance to pyrimethamine when expressed in *E. coli*. Though it did not drive differentiation in the PCA, *Poc dhfr-ts* haplotypes similarly showed the presence of a putative drug resistance haplotype (Ala15Ser + Ser58Arg)

in the Central and East African clones but not in West Africa, though our sample size for West Africa was generally smaller for both species (three and six isolates in *Pow* and *Poc*, respectively) (Supplemental Fig. 3B).

## Signatures of selection contain putative drug resistance loci, proteins involved in sexual stage differentiation, and antigenic targets

We calculated $nS_L$ and Tajima's D across the genomes to identify loci under directional and balancing selection, respectively. The $nS_L$ statistic is considered robust to the currently unknown recombination rates across the genomes of *P. ovale* species[43]. Genetic markers of interest within 10 kb of the top 0.5% absolute normalized $nS_L$ values that may be influenced by selective sweeps are listed in Tables 2, 3. Evidence of a selective sweep involving the putative bifunctional dihydrofolate reductase–thymidylate synthase (*dhfr-ts*) gene[41,42] was found in both *P. ovale* species (Fig. 5). Examination of extended haplotype homozygosity (EHH) at the selected variants show a large selective sweep in *Pow* spanning roughly 40 kb as well as close proximity of the *dhfr-ts* gene to the focal variant (Fig. 6A, B). In *Poc*, the positioning of the *dhfr-ts* gene lies at the edge of a smaller sweep (Fig. 6C, D). However, another putative marker of drug resistance, multidrug resistance-associated protein 2 (*mrp2*), was found in close proximity to one of the highest absolute normalized $nS_L$ values in *Poc* and lies near the center of a 40 kb sweep region on *Poc* chromosome 14 (Fig. 6E, F).

Top absolute $nS_L$ hits were also found near *ap2* transcription factor genes that regulate apicomplexan life cycle transitions, including sexual differentiation into gametocytes (*ap2-g*), and genes involved in sex-specific development of gametes, such as those coding male development protein 1 (*md1*) and cysteine-rich secretory protein (*crisp*)[44–46]. In *Pow*, four top $nS_L$ hits were found around genes encoding the dynein heavy and light chains, cytoskeleton components highly expressed in male gametes for motility and fusion with female gametes in the mosquito blood meal[47]. Top $nS_L$ hits were also found near cysteine repeat modular proteins 2 and 1 (*crmp2/1*) in *Poc* and *Pow*, respectively, proteins which may be involved in targeting sporozoites to the salivary glands in the mosquito prior to transmission[48,49].

Finally, genes encoding putative antigenic targets at the host-parasite interface, including merozoite surface protein 7 (*msp7*), merozoite surface protein 5 (*msp5*), early transcribed membrane protein (*etramp*), apical membrane antigen 1 (*ama1*), GPI-anchored micronemal antigen (*gama*), and 6-cysteine protein B9 (*6-cys*) may be under directional selection in both *P. ovale* species[50,51]. An orthologue of sporozoite protein essential for cell traversal 1 (*spect-1*), a

**Table 2 | Selection statistics and nearby genetic markers for loci with top normalized $n$S$_L$ and positive Tajima's $D$ values among monoclonal isolates of *Poc***

| Statistic | Value | Chr. | Location | Closest plausible genetic driver | Distance | Gene ID |
|---|---|---|---|---|---|---|
| $n$S$_L$ | 5.99 | 1 | 127,675 | HECT-type E3 ubiquitin ligase UT, putative (*ut*) | 1798 | PocGH01_01012400 |
| | −3.47 | 1 | 203,583 | cysteine-rich secretory protein, putative (*crisp*) | -4763 | PocGH01_01013600 |
| | −3.61 | 3 | 541,949 | cysteine repeat modular protein 2, putative (*crmp2*) | 6695 | PocGH01_03022100 |
| | −3.40 | 5 | 66,310 | early transcribed membrane protein, putative (*etramp*) | −790 | PocGH01_05011300 |
| | −3.38 | 5 | 107,531 | GPI-anchored micronemal antigen, putative (*gama*) | −3142 | PocGH01_05012100 |
| | −3.71 | 5 | 774,030 | bifunctional dihydrofolate reductase-thymidylate synthase, putative (*dhfr-ts*) | −9660 | PocGH01_05028400 |
| | −3.65 | 9 | 1,197,220 | apical membrane antigen 1, putative (*ama1*) | 605 | PocGH01_09039800 |
| | −3.55 | 12 | 703,579 | merozoite surface protein 7-like protein, putative (*msp7*) | 1983 | PocGH01_12027700 |
| | −4.03 | 12 | 2,760,110 | male development protein, putative (*md1*) | 0 | PocGH01_12076000 |
| | −4.01 | 14 | 1,607,839 | AP2 domain transcription factor, putative (*ap2-g*) | −4111 | PocGH01_14048300 |
| | −5.25 | 14 | 1,899,110 | ABC transporter C family member 2, putative (*mrp2*) | 7715 | PocGH01_14054800 |
| Tajima's $D$ | 2.66 | 7 | 1,152,840 | merozoite surface protein 1, putative (*msp1*) | 0 | PocGH01_07037900 |

Loci are described by chromosome (chr.), location on the chromosome in base pairs, and a statistical value in the top 0.5% across the genome. Nearby genetic markers were identified within 10,000 base pairs of these loci and are given alongside their gene ID and the distance of this marker to the reported locus (negative distance indicates upstream location). Source data are provided as a Source Data file.

**Table 3 | Selection statistics and nearby genetic markers for loci with top normalized $n$S$_L$ and positive Tajima's $D$ values among monoclonal isolates of *Pow***

| Statistic | Value | Chr. | Location | Closest plausible genetic driver | Distance | Gene ID |
|---|---|---|---|---|---|---|
| $n$S$_L$ | −4.61 | 2 | 537994 | dynein heavy chain, putative (*dhc*) | 0 | POWCR01_020017200 |
| | −4.79 | 2 | 554712 | transcription factor with AP2 domain(s) (*apiap2*) | 7318 | POWCR01_020017600 |
| | −5.08 | 3 | 554677 | dynein heavy chain, putative (*dhc*) | 2069 | POWCR01_030017700 |
| | −4.99 | 4 | 603847 | merozoite surface protein 5, putative (*msp5*) | −5083 | POWCR01_040018700 |
| | −4.89 | 5 | 846693 | bifunctional dihydrofolate reductase-thymidylate synthase, putative (*dhfr-ts*) | −953 | POWCR01_050023500 |
| | −4.69 | 7 | 456511 | cysteine repeat modular protein 1, putative (*crmp1*) | −6204 | POWCR01_070013600 |
| | −5.45 | 7 | 1080681 | dynein light chain, putative (*dhc*) | 8560 | POWCR01_070029300 |
| | −5.22 | 8 | 1135323 | 6-cysteine protein B9 (*6-cys*) | 4768 | POWCR01_080029500 |
| | −4.39 | 9 | 792065 | dynein heavy chain, putative (*dhc*) | 0 | POWCR01_090024400 |
| | −4.93 | 12 | 525249 | sporozoite protein essential for cell traversal, putative (*spect1*) | −5947 | POWCR01_120016200 |
| | −4.10 | 13 | 177064 | *Plasmodium* interspersed repeat protein (*pir*) | −5114 | POWCR01_130008000 |
| | −4.10 | 13 | 177064 | early transcribed membrane protein, putative (*etramp*) | 6011 | POWCR01_130008100 |
| Tajima's $D$ | 2.55 | 1 | 31910 | conserved Plasmodium protein, unknown function | 0 | POWCR01_010005700 |
| | 2.58 | 1 | 128700 | serine/threonine protein kinase, putative | 0 | POWCR01_070032200 |
| | 2.13 | 7 | 1184250 | merozoite surface protein 1, putative (*msp1*) | 0 | POWCR01_080035700 |
| | 2.81 | 12 | 235390 | stromal-processing peptidase, putative | 0 | POWCR01_130008300 |

Loci are described by chromosome (chr.), location on the chromosome in base pairs, and a statistical value in the top 0.5% of absolute values across the genome. Nearby genetic markers were identified within 10,000 base pairs of these loci and are given alongside their gene ID and the distance of this marker to the reported locus (negative distance indicates upstream location). Source data are provided as a Source Data file.

protein necessary for liver cell invasion that has been investigated as a potential vaccine target[52], was also among the top hits in *Pow*.

Overall, Tajima's D in both species exhibited a negative skew across the genome with an average value of −1.06 for *Poc* and −0.78 for *Pow*. This may suggest population expansion following a bottleneck or weak directional selection (Fig. 7). Among the loci with positive values in the top 0.5% of absolute Tajima's D hits, the antigenic marker merozoite surface protein 1 (MSP1) was identified as a probable target of balancing or diversifying selection in both *P. ovale* species (Tables 2, 3).

## Discussion

We present a comprehensive population genomic study of both *P. ovale* species within sub-Saharan Africa. Our study comprises 21 *Poc* and 24 *Pow* isolates selected from 11 studies, including both febrile and asymptomatic cases. Genome-wide analysis reveals differences in

nucleotide diversity between *P. ovale* species, but similarity in their low complexity of infection, geographic relatedness, and signatures of selection. Our analysis was performed using genomic enrichment methods specifically designed to enable robust coverage and analysis with the 2017 reference genomes of *P. ovale curtisi* (from Ghana) and *P. ovale wallikeri* (from Cameroon), which were the available references at the time of the study[28].

Compared to the "classic" *P. ovale curtisi* species, we observed significantly lower nucleotide diversity across orthologous genes among geographically matched *P. ovale wallikeri* isolates ($2.5 \times 10^{-4}$ for *Poc* and $1.8 \times 10^{-4}$ for *Pow*, respectively). Our estimate for *Poc* is concordant with the genome-wide diversity calculated among six Central African *Poc* isolates[15] as well as that derived from RNA expression data among four parasite samples from Mali[53]. However, our *Pow* estimate was substantially lower than the genome-wide estimate reported by

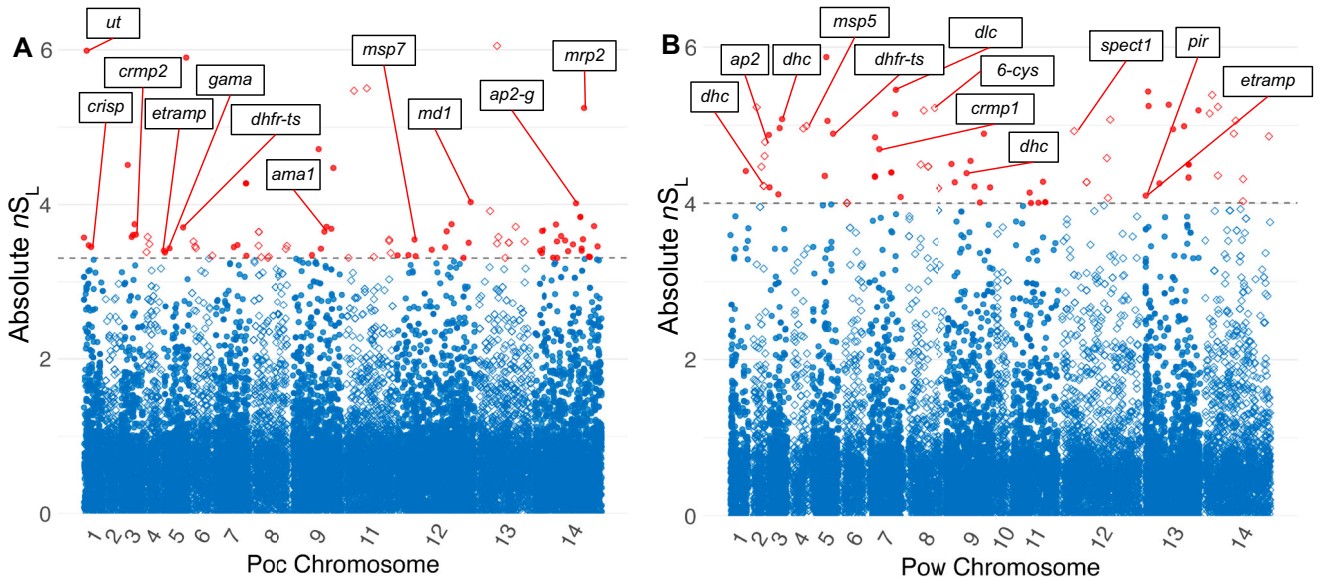

**Fig. 5 | Absolute $n$S$_L$ across the *Poc* and *Pow* genome.** Absolute $n$S$_L$ of 19,205 variants among monoclonal isolates across the *Poc* genome (**A**) and 15,744 variants among monoclonal isolates across the *Pow* genome (**B**). Individual loci are depicted using alternating shapes between chromosomes for legibility. The dotted line and red color denote the top 0.5% of loci. Putative genetic markers of note within 10,000 bases of these loci are labeled, including ubiquitin transferase (*ut*), cysteine-rich secretory protein (*crisp*), cysteine repeat modular protein 2 (*crmp2*), early transcribed membrane protein (*etramp*), GPI-anchored micronemal antigen (*gama*), dihydrofolate reductase-thymidylate synthase (*dhfr-ts*), apical membrane antigen (*ama1*), merozoite surface protein 7-like protein (*msp7*), male development protein 1 (*md1*), AP2 domain transcription factor G (*ap2-g*), multidrug resistance-associated protein 2 (*mrp2*), dynein heavy chain (*dhc*), AP2 domain transcription factors (*ap2*), merozoite surface protein 5 (*msp5*), cysteine repeat modular protein 1 (*crmp1*), dynein light chain (*dlc*), 6-cysteine protein (*6-cys*), sporozoite protein essential for cell traversal 1 (*spect1*), *Plasmodium* interspersed repeat protein (*pir*), and early transcribed membrane protein (*etramp*). Source data are provided as a Source Data file.

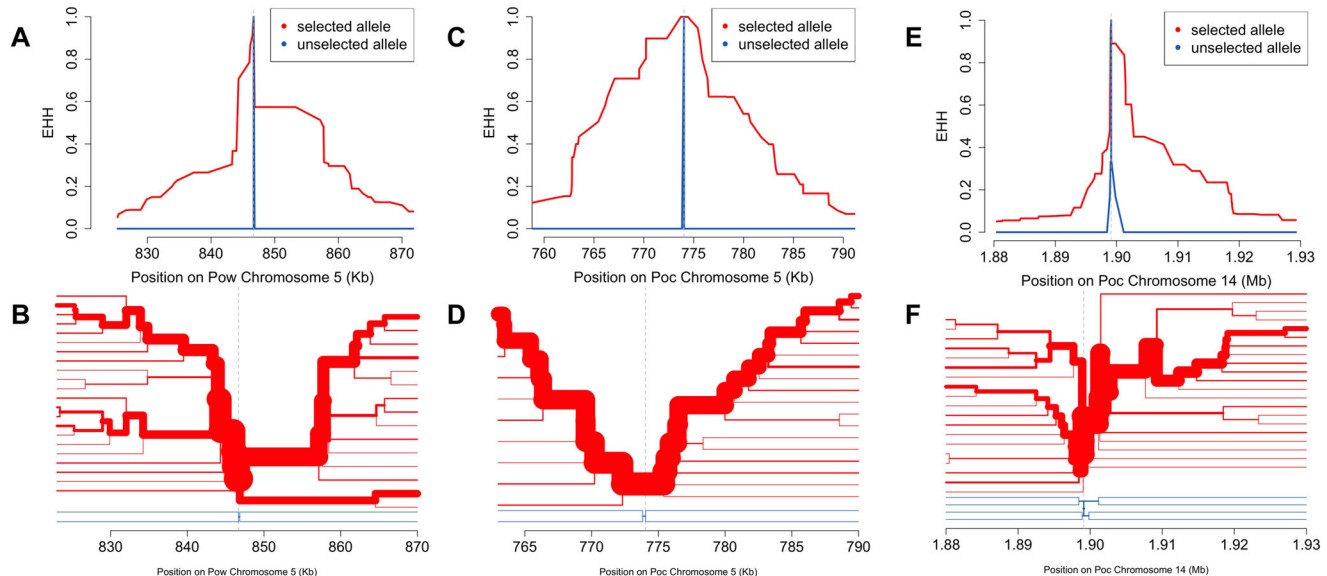

**Fig. 6 | Extended haplotype homozygosity and haplotype bifurcation at selected variants.** Extended haplotype homozygosity (EHH) and haplotype bifurcation among monoclonal isolates at selected variants near the *Pow dhfr-ts* gene (**A**, **B**), the *Poc dhfr-ts* gene (**C**, **D**), and the *Poc mrp2* gene (**E**, **F**). EHH and haplotype bifurcation show selective sweeps spanning ~30, ~40, and ~30 kb, respectively, with lineage breakdown occurring first among the unselected allele haplotypes (blue) and then in the selected allele haplotypes (red) as the distance from the focal variant increases.

Higgins et al. ($3.4 \times 10^{-4}$), despite our inclusion of their samples alongside additional Central and East African isolates. Our lower estimate may reflect the exclusion of higher-diversity intergenic regions, though we also found lower genome-wide and intergenic SNP density in *Pow* compared to *Poc*. Relatively low nucleotide diversity in *Pow* may indicate reduced effective population size, increased inbreeding, or a population bottleneck in the time since *Poc* and *Pow* diverged between 1.3 and 20.3 million years ago[14,28]. More recent population growth in both species is also suggested by the predominantly negative distribution of Tajima's D values across their protein-coding genes, a finding that can indicate population expansion following a bottleneck[54]. The high ratio of nonsynonymous-to-synonymous

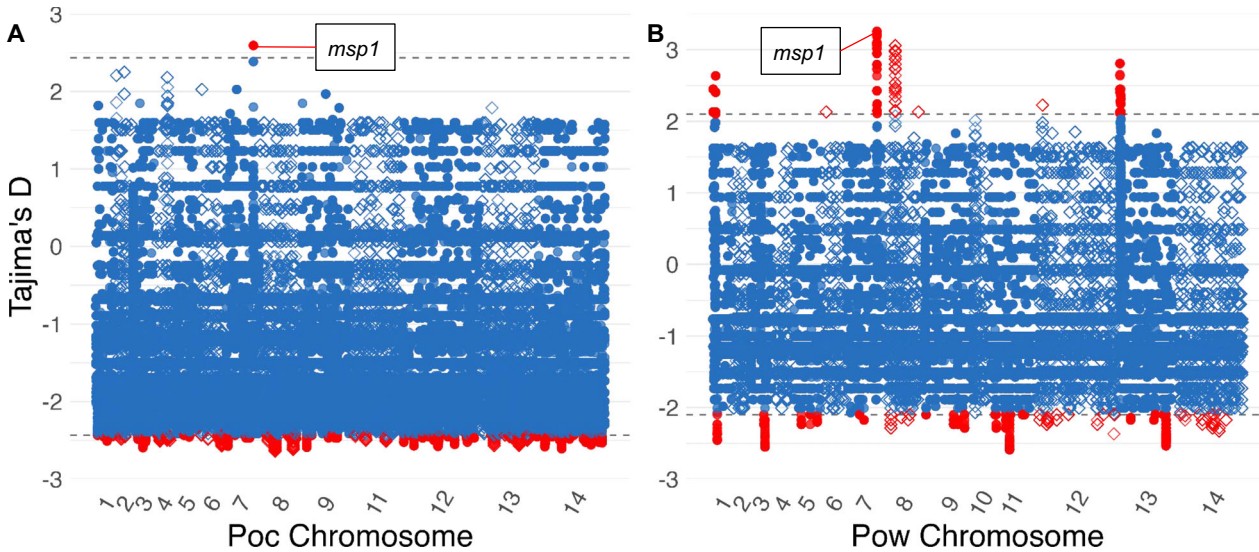

**Fig. 7 | Tajima's D across the *Poc* and *Pow* genomes.** Tajima's D in 399,355 and 297,286 300 bp windows in genes across the *Poc* (**A**) and *Pow* (**B**) genomes among monoclonal isolates. Individual windows are depicted using alternating shapes between chromosomes for legibility. The dotted line and red color denote the top 0.5% of loci. For both species, the top locus with a positive Tajima's D was located inside the merozoite surface protein 1 (*msp1*) gene. Source data are provided as a Source Data file.

substitutions among these protein-coding genes (2.5 for *Pow*, 1.5 for *Poc*) is similar to that seen in *P. falciparum* and *P. vivax*[55,56]. This finding may represent diversifying selection on proteins across either *P. ovale* genome, enabling maintenance of nonsynonymous substitutions, or inflation of dN/dS ratios observed among *Plasmodium* parasites due to the impact of the malaria life cycle on allele frequencies[57]. Further analysis of subpopulations of each parasite species could help to elucidate the factors driving the observed difference in genomic diversity, such as by determining whether *Pow* isolates from Asia have similarly low nucleotide diversity or if this finding is specific to Africa.

The observed predominance of monoclonal isolates among both *P. ovale* species is consistent with low within-sample haplotypic diversity seen in previous investigations of African *P. ovale* isolates by genome-wide RNA sequencing and amplicon sequencing[53,58]. The low complexity of *P. ovale* infections may result from efficient clonal transmission[32] and/or lower transmission overall, limiting vector uptake of multiple parasite clones from either the same or different infected individuals. This low complexity is expected to limit opportunities for genetic recombination within mosquito vectors, though multiple-clone infections were identified in Kinshasa, a region with overall higher malaria endemicity and transmission intensity[39].

Genomic signatures of selection within both *P. ovale* species highlighted the importance of antimalarials, host-vector life cycle transitions, and human immunity as evolutionary pressures impacting parasite survival. *P. ovale* infections are frequently subclinical and go untreated[59], but likely still face substantial drug exposure from widely used antimalarials prescribed for *P. falciparum*[22]. Additionally, malaria prophylaxis using sulfadoxine-pyrimethamine, such as intermittent preventive therapy in pregnancy (IPTp) and seasonal malaria chemo-prevention for infants and schoolchildren (SMC), may be applying drug pressure on *P. ovale* parasite populations[1]. Selective sweeps in *dhfr-ts*, a gene implicated in resistance to pyrimethamine, have been documented in both *Poc* and *Pow*, and certain mutant alleles were found to confer pyrimethamine resistance when expressed in *E. coli*[24,25]. In our dataset, sweeps near the *dhfr-ts* genes were among the strongest signals of directional selection in both *P. ovale* species, especially in *Pow*, possibly representing drug pressure influencing parasite survival. We also found the putative pyrimethamine resistance *Pow dhfr-ts* haplotype Phe57Leu + Ser58Arg to be a major contributor to principal component 2 of *Pow* (representing the North-South axis);

the resistant haplotype composed 36% (9/25) of our *Pow* haplotypes, with representation in Central and East Africa but no detection in West Africa. Another putative pyrimethamine resistance haplotype (Ala15-Ser + Ser58Arg) in *Poc* was similarly detected in East and Central Africa, though not in West Africa[25]. Functional evaluation of different alleles to determine their capacity to confer drug resistance, as well as monitoring of these alleles across the parasite populations over time, will further clarify how interventions targeting *P. falciparum* may be simultaneously rendering *P. ovale* parasites harder to control.

Finally, strong signals of balancing or diversifying selection were observed in both species within their genes encoding merozoite surface protein 1 (MSP1), the orthologues of the predominant antigen on blood-stage *P. falciparum* parasites that has been shown to induce protective immunity in some studies[60]. Diversifying selection on MSP1 has been documented in *P. falciparum*[61] as well as in focused analysis among African *P. ovale* infections imported to China[62]. Our dataset provides even stronger evidence for diversifying selection at this site, as the *msp1* gene showed the single highest Tajima's D value across all genes in both *P. ovale* species. Such immune responses may also play a role in modulating relapse potential[63].

This study has several limitations. While it represents the largest genome-wide examination of the genomic composition of both *P. ovale curtisi* and *P. ovale wallikeri* to date, the sample size for both species is nonetheless small and limits the power to detect clustering of isolates, infer population demography, and detect selection. The geographic coverage of the isolates employed differs between the 13 countries represented, and isolates from Northern or Southern Africa were not available. Whole-genome enrichment methods also differed among isolates; 21 isolates employed hybrid capture, 15 used selective whole-genome amplification, and nine relied on leukodepletion. The two former methods may have induced amplification bias, whereas leukodepletion does not amplify *P. ovale* DNA and, therefore, may reduce the power to identify rare variants in those isolates. Disparate average read depth between the three species (44.1, 95.7, and 147.5 for *Poc*, *Pow*, and *Pf*, respectively) may also have differentially impacted our ability to detect polyclonal infections, but the low complexity of infection found in both *P. ovale* species should be robust given the satisfactory sequencing depth overall. The source studies also differ in whether samples were derived from asymptomatic carriers (n = 11) or febrile patients (n = 34). Sample sizes were too small to analyze these

populations separately. Finally, hybrid capture baits designed using the incomplete PowCR01 reference genome led to incomplete coverage of *Poc* chromosome 10, which was excluded from analyses. Unfortunately, newly assembled regions of *Pow* chromosome 10 were not available during the analysis. The hybrid capture approach also did not enable enrichment and analysis of loci in the mitochondrial and apicoplast genomes, which were excluded from analysis. We do not expect these exclusions to systematically bias the estimation of nucleotide diversity or complexity of infection, though it does prevent us from evaluating excluded loci (including *mdr1*, *msp3*, and *msp8*) for genomic signatures of selection. The availability of selective whole genome amplification protocols now provides a less expensive approach to targeted DNA enrichment for *P. ovale* spp. that does not rely on the specific design of hybrid capture baits[27].

This study provides a comparative genomic analysis of the two *Plasmodium ovale* species sympatrically circulating in sub-Saharan Africa and presents new evidence of selective pressures on genes related to drug response, sexual differentiation, and immune evasion. Further population genomic studies of *Poc* and *Pow* should employ a larger selection of isolates from a greater geographic range, especially including Asia, and take advantage of new reference genome assemblies to build on these insights[15]. Functional investigation into the genes showing signatures of selection, including via orthologue replacement in closely related *Plasmodium* species[64,65], is also an exciting new strategy in substantiating the biological relevance of key loci, with implications for transmission prevention, treatment strategies, and vaccine development for *P. ovale* spp. Finally, cataloging genome-wide diversity facilitates the design of targeted genotyping methods that can efficiently characterize the epidemiology of these understudied parasite species[66]. Combining these approaches to better evaluate *P. ovale* parasite relatedness, transmission, and relapse patterns can help to improve the impact of current malaria control strategies on all human-infecting malaria species[36,67,68].

## Methods

### Ethics
This analysis was considered non-human subjects research by the University of North Carolina. The appropriate collection of samples, authorization for use in genomic studies, and IRBs involved are summarized in the original studies outlined in Supplemental Table 1.

### Sample selection
Clinical isolates in the form of dried blood spots or leukodepleted blood were drawn from six studies shown in Table 1, including studies involving both asymptomatic persons and febrile patients across four countries. Across these studies, participants were screened for the presence of *P. ovale* spp. infection by a real-time polymerase chain reaction (qPCR) assay targeting the *po18S* rRNA gene[22]. Among 282 isolates with a *po18S* Ct value under 40, a species-specific (*Poc* and *Pow*) 18S rRNA qPCR assay was employed to determine the ovale species present[69]. Candidates were selected from isolates with only one species detected or mixed infections in which one species predominated by ≥3 Ct (corresponding to approximately 8 times as much DNA). Samples were also screened for the presence of *P. falciparum* coinfection using a qPCR assay for the *pfldh* or *pf18S* rRNA gene[31,39]. Ultimately, samples from 25 individuals were selected for whole-genome sequencing based on higher-density *P. ovale* infection, lack of or lower-density *P. falciparum* coinfection, and balance of ovale species and geographic diversity across the sample set. Characteristics of these 25 samples, and an additional 20 samples from four previously published studies[15,28,36,37], are shown in Supplemental Table 1.

### Library preparation and sequencing
DNA extracted from dried blood spots using a Chelex protocol[70] was sheared to 300 bp using a LE220R-plus Covaris Sonicator (Covaris, Woburn, MA). Fragment size was checked with an Agilent TapeStation 4150 (Agilent, Santa Clara, CA), and DNA concentrations were tested using a Qubit Flex fluorometer (Thermo Fisher Scientific, Waltham, MA). Isolates were then prepared for sequencing using the KAPA Hyperprep kit (Kapa Biosystems, Woburn, MA). Four Tanzanian DNA isolates extracted from blood that had been leukocyte-depleted by CF11 filtration at the time of collection[35] were directly incorporated into sequencing libraries. The remaining 21 isolates derived from dried blood spots were additionally processed using a custom-designed hybrid capture protocol to enrich for ovale DNA via thousands of RNA probes specifically designed to amplify *P. ovale* DNA without binding to human DNA (Twist Bioscience, San Francisco, CA, USA). Hybrid-capture probes were designed first for the *P. ovale wallikeri* genome (PowCR01), with unique probes added for the *P. ovale curtisi* genome (PocGH01) at any sites that differed by more than 10% of bases[28]. Since the reference genome for chromosome 10 is significantly smaller for *Poc* compared to *Pow* (roughly 1300 vs 470 kb, respectively), this bait design approach led to a lack of baits covering 63% of *Poc* chromosome 10. Chromosome 10 was therefore excluded from the analysis of *Poc* isolates. Captures were performed with four samples per capture. After preliminary sequencing on a Miseq Nano flow cell (Illumina, San Diego, CA, USA), libraries were sequenced on the Novaseq 6000 S Prime (Illumina, San Diego, CA, USA) sequencing system with 150 bp paired-end chemistry. Samples from Joste et al. and Higgins et al. included in our analysis were enriched using selective whole genome amplification (sWGA), employing sets of five to ten primers designed to preferentially amplify the PowCR01 and PocGH01 genomes over human background DNA[71].

### Sequencing data alignment and variant calling
Data processing and analysis were performed in the bash environment using a Python-based *snakemake* v7.24.2 wrapper for pipeline construction, automation, and reproducibility[72]. Raw sequencing reads were trimmed of Illumina adapters using *Trimmomatic* v0.36[73] before the quality of the reads was evaluated with *fastQC* v0.11.9[74]. A "dual" reference genome was then produced for each *P. ovale* species by concatenating the reference genome of that ovale species (*P. ovale curtisi*: PocGH01; *P. ovale wallikeri*: PowCR01) to the *P. falciparum* strain *Pf3D7* reference genome[28,75]. Paired reads from each isolate were aligned to their species' corresponding dual reference genome using *bwa-mem2* v2.2.1[76] before each alignment was sorted and given read group information using *picard* v2.26.11[77] and deduplicated via *GATK* v4.4.0.0[78]. *Samtools* v1.17 was then used to select for reads that aligned to the *P. ovale* portion of the dual reference genome rather than that of *P. falciparum*, thus discarding reads from contaminating *P. falciparum* DNA present in some isolates[79]. Resulting alignments of read that preferentially mapped to *P. ovale* were soft-clipped to reference genome edges and cleaned of unmapped reads using *GATK*, after which mapping proportion and coverage of the ovale reference genome were calculated by *samtools* and *bedtools* v2.30[80].

Variant calling from aligned reads across each ovale species genome was also performed using the *GATK* best practices pipeline[78]. In the resulting callset, variants were masked via *vcftools*[81] if they fell outside of the 14 chromosomes of the reference genome or were within the following expanded gene families (which were masked to reduce sequence error in repetitive genomic elements): *Plasmodium* interspersed repeat protein (*pir*), surfin-related subtelomeric protein 1 (*stp1*), early transcribed membrane protein (*etramp*), tryptophan-rich antigen (*tra*), *Plasmodium* exported protein (*phist*), reticulocyte binding protein (*rbp*), ookinete surface protein (*osp*), 6-cysteine protein, KELT protein, and pm-fam-a protein[37]. Tandem repeats across each species' genome were identified and masked using Tandem Repeat

Finder v4.09.1[82]. *GATK* hard filtering was then used to remove variants with poor quality metrics using the following filter thresholds: quality by depth <3, Fisher strand bias >50, strand odds ratio >3, mapping quality >50, mapping quality rank sum <−2.5, read position rank sum < −3. Call sets were limited to biallelic single nucleotide polymorphisms (SNPs) that were present in at least 80% of individuals. SNP density across the entire genome and within specific functional regions of each genome were calculated using custom scripts (see Data availability). *SNPeff* v4.3 was used to annotate individual variants and determine the ratio of nonsynonymous-to-synonymous mutations[28,83].

## Selection of *P. falciparum* comparison dataset

Co-endemic *P. falciparum* samples were drawn from the Pf6 dataset[38]. Of 20,705 total *P. falciparum* isolates from around the globe, 2077 came from the same or nearby geographic locations as the source studies of *P. ovale* isolates described above. Thirty-two *P. falciparum* samples with over 85% base callability were randomly selected in order to have at least one co-endemic *P. falciparum* isolate for each *P. ovale curtisi* or *ovale wallikeri* isolate (Supplemental Table 2). Variant call sets for these *P. falciparum* samples were limited to the falciparum core genome[75], quality filtered by Variant Quality Score Recalibration[78], and restricted to sites present in at least 80% of individuals.

## Complexity of infection (COI) calculation

Variant call sets were limited to variants with a minor allele frequency greater than or equal to 5%. The McCOILR R package was employed to run THEREALMcCOIL on each sample set using 1000 Markov chain Monte Carlo iterations, 100 burn-in iterations, a maximum COI of 25, a minimum number of sites for a sample to be included of 10, and a minimum number of samples for a site to be included of 10 (analysis was run on 21 and 24 *P. ovale curtisi* and *ovale wallikeri* isolates, respectively)[84]. This yielded 1000 estimates of the complexity of infection, or the number of unique parasite clones found in each sample, and median COI per isolate was reported. COI distributions among *P. falciparum* isolates were calculated in the same manner. COI distributions among the three species were compared using a Kruskal–Wallis test, with Dunn's multiple comparisons test employed for specific pairwise comparisons[85].

To investigate the distribution of heterozygous SNPs in samples that were determined to be polyclonal, the variant set was filtered to sites with >10% minor allele frequency across the whole population and a within-sample minor allele frequency >50% for at least one sample. Then, for each polyclonal sample, the remaining sites were filtered to those with a within-sample minor allele frequency greater than or equal to 5%.

## Principal components analysis

After limiting to monoclonal samples (20 *Poc*, 23 *Pow*), variant call sets were filtered to remove sites with a minor allele frequency of less than 5% using *vcftools* v0.1.15[81]. Variants were processed using PLINK v1.90b6.21, first by pruning variants within windows of 50 variants between which the $R^2$ value exceeded 0.3 (windows were shifted by steps of five variants for each pruning step)[86]. Then, principal components analysis was performed in PLINK, including extraction of the weights by variant. Data visualizations were performed in R using the *ggplot2* v3.4.4 package[87]. ADMIXTURE v1.3.0 was employed to test for significant clustering among samples within each species[88].

## Nucleotide diversity calculation

For both ovale species and *P. falciparum*, the OrthoMCL database was searched for orthologous gene sets in which there was only one copy of each ortholog per genome[89]. Sets were removed from the total pool of orthologous genes if they were partially or totally masked by the multigene family or extrachromosomal masks within an individual species, or if any orthologs did not have ≥5x coverage of the gene at least every 10 bases in at least 60% of isolates in each species. Among the remaining ortholog sets with sufficient coverage among all species, nucleotide diversity (π) was calculated across the monoclonal samples of each species for each ortholog in the set using *vcftools* v0.1.15[81]. The *P. falciparum* isolate set was composed of 19 monoclonal isolates selected to ensure one geographically matched *Pf* sample for each *P. ovale* sample (Supplemental Table 2). Variant call sets in this analysis did not employ a minor allele frequency filter. Nucleotide diversity was compared between species among the remaining sets of orthologs using repeated measures ANOVA; pairwise comparisons were examined with two-tailed Tukey's multiple comparisons test. To mitigate potential bias introduced by differing geographic coverage between the two ovale species (and by differences with the *P. falciparum* genome), nucleotide diversity was also calculated among a set of orthologue sets (identified among one-to-one *Poc-Pow* orthologues with strong coverage as detailed above) in geographically matched *Poc* and *Pow* samples (n = 11 each, Supplemental Table 3) and compared using Wilcoxon's matched-pair signed rank test[90].

## Identification of signatures of selection

Variant call sets were limited to monoclonal isolates and sites which had no missingness in any samples. After employing the default minor allele frequency filter of >5%, *selscan* v1.2.0 was used to calculate $nS_L$ (a metric of directional selection) for all remaining variants in each species and the *norm* function used to normalize values in allele frequency bins[91]. $nS_L$ was chosen as it is more robust to the currently unknown recombination rates across either *P. ovale* genome than metrics like iHS and because it can evaluate selection within a single population of organisms. Loci with an $nS_L$ in the most extreme 0.5% were investigated in PlasmoDB for proximity to genes of interest within 10 kb[89]. Extended haplotype homozygosity (EHH) was calculated at select sites using *rehh*[92]. Tajima's D (a metric of possible directional and balancing selection) was calculated in 300 base pair sliding windows shifted by 10 bp using all non-missing variants across all known genes in each genome using *vcf-kit* v0.2.9[93,94]. Variants with positive Tajima's D values in the top 0.5% of absolute values were investigated in PlasmoDB.

## Reporting summary

Further information on research design is available in the Nature Portfolio Reporting Summary linked to this article.

## Data availability

All new sequence data are available at NCBI SRA: BioProject ID PRJNA1092086. Public data used include the contents of the following projects/studies: European Nucleotide Archive Study accession number PRJEB51041, Run accession numbers: ERR10738334, ERR10738339, ERR10738341, and ERR10738346. SRA Study accession number PRJEB13344 [https://www.ncbi.nlm.nih.gov/bioproject/PRJEB13344], Run accession numbers: ERR1739852, and ERR1739853. SRA Study accession number PRJEB12679, Run accession numbers: ERR1428159, ERR1254542, and ERR1254543. SRA Study accession number PRJNA1015456, Run accession numbers: SRR26037552, SRR26037551, SRR26037550, SRR26037549, SRR26037548, SRR26037546, SRR26037545, SRR26037544, SRR26037543, SRR26037542, and SRR26037541. Additional Run accession numbers: ERR404145. ERR404154. ERR377533. ERR404191. ERR404207. ERR1045266. ERR1045267. ERR676479. ERR1106575. ERR1106579. ERR1106586. ERR1106587. ERR1106590. ERR449901. ERR449903. ERR405238. ERR405244. ERR666939. ERR562889. ERR636018 [https://www.ncbi.nlm.nih.gov/sra/?term=ERR562889]. ERR912913 [https://www.ncbi.nlm.nih.gov/sra/?term=ERR636018]. ERR1514567. ERR1045287. ERR1172616. ERR1172593. ERR1172615. ERR1172608. ERR059405. ERR045598. ERR666937. ERR580480. ERR701763 Source data are provided with this paper.

## Code availability

All code for processing and analysis of samples is available at: Carey-Ewend, K., Population genomics of Plasmodium ovale species in sub-Saharan Africa, https://github.com/bailey-lab/Po_popgen_snakemake/tree/main/final, https://doi.org/10.5281/zenodo.14026786[95].

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

## Acknowledgements

We thank all the participants of the parent studies for providing the biological material for the study. The following reagents were obtained through BEI Resources, NIAID, NIH: (1) *P. falciparum*, strain 3D7A, MRA-151, contributed by David Walliker and (2) Diagnostic plasmid containing the small subunit ribosomal RNA gene (18S) from Plasmodium ovale, MRA-180, contributed by Peter A. Zimmerman. We also thank Dr. Parul Johri for her feedback on the final manuscript and Dr. Kevin Wamae for his support in bioinformatic pipeline construction. This study was funded by the National Institute for Allergy and Infectious Diseases, National Institutes of Health (R01AI137395 and R21AI152260 to J.T.L., R21AI148579 to J.B.P. and J.T.L., R01AI65537 to J.J.J., K24AI134990 to J.J.J., R01AI129812 to A.T., R01AI132547 to J.J.J., R01TW010870 to J.J.J., T32AI070114 supporting K.C.-E.), the Fogarty Center, the Global Fund to Fight AIDS, Tuberculosis, and Malaria, and the Bill and Melinda Gates Foundation (Inv. No. 002202 to D.I., J.J.J., and J.A.B.). Under the grant conditions of the Bill and Melinda Gates Foundation, a Creative Commons Attribution 4.0 Generic License has already been assigned to the Author Accepted Manuscript version that might arise from this submission.

## Author contributions

K.C.-E., J.T.L., J.J.J., and J.B.P. conceived and designed the study. S.F., F.P., K.M., O.A., D.I., I.A., B.B., B.N., A.K., and A.T. contributed to clinical sample collection. K.C.E., Z.P.H., M.M., C.H., and C.G. conducted laboratory work. K.C.-E., Z.P.H., K.N., W.H., F.A., and A.S. conducted data analysis. K.C.-E. drafted the first version of the manuscript. K.C.-E., J.T.L., J.J.J., Z.P.H., J.B., C.S., and J.B.P. edited and finalized the manuscript.

## Competing interests

J.B.P. reports research support from Gilead Sciences, non-financial support from Abbott Laboratories, and consulting for Zymeron Corporation, all outside the scope of this study. The remaining authors declare no competing interests. The findings and conclusions in this report are those of the author(s) and do not necessarily represent the official position of the Bill and Melinda Gates Foundation or other funders.
