## [Peer Review File · Nature Communications]

REVIEWER COMMENTS

Reviewer #1 (Remarks to the Author):

Carey-Ewend and colleagues presented a population genomics study on two sympatric species of human malaria parasites *P. ovale wallikeri* or 'Pow' (recently designated as *P. ovalewallikeri*) and *P. ovale curtisi* or 'Poc' (recently designated as *P. ovalecurtisi*). They sequenced the genomes of 25 isolates of Poc and Pow (13 isolates of Poc and 12 isolates of Pow) from central and east Africa (Ethiopia, the DRC, Tanzania, and Cameroon). They included four additional previously published genomes from west and central Africa – thus making the total number of analysed genomes 29. The authors used hybrid capture or leukodepletion to enrich Pow and Poc DNA material and performed whole genome sequencing (WGS). The authors used a comprehensive array of population genomics tools to determine average nucleotide diversity, signatures of balancing selection and selective sweeps at specific genomic loci within their study population. The authors observed evidence for balancing selection in major known malaria antigens such as *ama1* and *msh1* and a distinct signal of a selective sweep in the *dhfr-ts* gene implicated in resistance to pyrimethamine. The manuscript utilises state-of-the-art methodologies in population genomics, and in most cases, the results reconfirm data already shown for both sympatric malaria species in previous publications. Overall, the authors have analysed the data well and presented their results in nicely presented figures and tables. While I enjoyed reading the manuscript, several questions came to my mind about the novelty of the findings in light of what is already known about these parasites at the population level. Some publicly available data sources have not been included in the analysis, and some claims need a larger sample size from wider sources to be substantiated.

Major Points:

1. I am concerned that the conclusions are drawn based on relatively few genomes (16 Pow and 13 Poc) to be highly informative of sub-Saharan Africa. Only African samples have been considered, and both Pow and Poc are present in other regions of the world. They would need to be included in the analysis to make claims of “comprehensiveness” and for robust interpretation. Indeed, the authors allude to this limitation about the sample size and the study limitations (Line 368). I am wondering why a few other publicly available WGS datasets on both Pow and Poc have not been included in the study that were generated as part of the first two studies describing the genomes of Pow and Poc (PMID: 27392654; PMID: 28117441) in 2016-17 and more recently (2024) in PMID: 38360879. Moreover, I would have liked to see parasite samples from a wider region in the world included in such a study, including parasites from the Indian subcontinent, SE Asia, and other parts of Africa. While establishing access to such globally representative samples may be challenging and time-consuming, I would have liked to see more samples, at least from the African continent, included in the study to make the observations robust and “comprehensive”.
2. Perhaps one of the most interesting things is the clear signal of the selective sweep on the *dhfr-ts* locus; however, that has been reported twice before. In this regard, the authors are missing a major reference (PMID: 38761813), who also performed a population genetics analysis of *dhfr* mutations in 518 *P. ovale* spp samples (314 Poc and 204 Pow) from Benin, Gabon, and Kenya and found selective sweeps in the *dhfr* locus (as did Chen et al, who they do reference).

3. I am also wondering if the Pow and Poc genomes (Poc221 and Pow 222) described in the Higgins et al. study (reference 15) could be used as a reference to capture better coverage and, hence, SNP statistics.

4. In the discussion (line 386), the authors claimed that they ‘shed light on the evolutionary history of the two Plasmodium ovale species’. This is not entirely true. What ‘evolutionary history’ have they uncovered here with 29 isolates representing two sympatric species? This is somewhat hyperbolic, as this study does not do this.

5. Nomenclature: I would like to draw attention to the recently revised nomenclature of these two sympatric species as Plasmodium ovalewallikeri and P. ovalecurtisi (PMID: 38040603) and the subsequent debate associated with it (PMID: 38160179; PMID: 38272740). The authors mention that this nomenclature is ‘currently evolving’ (Line 82) but do not refer to these recent publications. I prefer the revised binomial nomenclature proposed by Snounou et al. (PMID: 38040603). However, I leave it to the editor to provide appropriate guidance.

Minor Points:

1. In line 140, the authors refer to Chromosome 10 of Pow as being underrepresented. Is it still the case that Chromosome 10 of Pow continues to be underrepresented despite all the genome sequencing and assembly efforts using a combination of various 4th-generation long-read sequencing technologies?

2. Line 281: Figure 6C refers to Chromosome 14, yet the text gives Chromosome 13. Which is correct?

Line 297: Is this study really ‘comprehensive’?

Line 306: How confident are the authors of this? What evidence do they have for this claim?

Line 319: The timing of the divergence of these two species is questionable but is presented here as a fact.

Line 353 How, exactly, do the authors propose that “Functional evaluation of different 353 alleles to determine their capacity to confer drug resistance, as well as monitoring of these alleles across the parasite populations over time”, will be “necessary to ensure that P. ovale do not develop resistance to prophylactic malaria treatment.” P. ovale spp. are already developing resistance.

Reviewer #2 (Remarks to the Author):

The study by Ewend and colleagues provides new genomic resources (specifically new data) on the relatively neglected Plasmodium species, P ovale curtisi (Poc) and P ovale wallikeri (Pow). This is important for the malaria community as many countries experiencing a decline in P. falciparum are seeing a relative rise in the proportion of other malaria-causing species such as P ovale curtisi and wallikeri. Understanding the genetic epidemiology of these neglected species and detecting adaptations of potential public health concern is therefore critical and may be aided by genomic data such as

provided here. However, there are some critical limitations in the genomic data such as the low capture across the entirety of chromosome 10.

Other genomic studies of *P. ovale* have been conducted, as transparently referenced by the authors (ref 15, 22-24). The novelty of the current study is in being the first study to compare Poc against Pow with genome-wide measures. This enabled the authors to derive insights such as the apparent greater diversity in Poc relative to Pow. However, as the authors acknowledge, some of these inferences are moderately uncertain as the small sample size and opportunistic sampling render the comparisons susceptible to bias. As the authors also raise, the inferences on complexity of infection are also potentially vulnerable to bias relating to sample processing methods; indeed the loss of coverage across chromosome 10 highlights the potential impacts here.

I commend the authors in at least being very transparent about the study limitations so that readers can interpret the results appropriately. However, there are a few areas where further work could enhance both the comprehension of the study and the interpretation of some of the results. Suggestions are provided below;

1. One of the main limitations is in the small sample size (good for ovale but small nonetheless!) and poor representation of some geographic regions. The authors note that the new study (not available when they conducted their analysis) by Higgins et al adds new geographic representation. I expect they may already be working on this but if not, I would suggest that they integrate the data from Higgins to improve the comprehension of their study.

2. The loss of chromosome 10 from the hybrid capture samples is a shame but can't be changed at this point. I was a little concerned that in certain places the text still seems to promote hybrid capture over sWGA despite this problem. I would recommend being clearer in the messaging to other researchers who might want to conduct ovale genomics on the problems with the hybrid capture. For example, the authors don't mention it, but key drug resistance targets such as *mdr1* (and perhaps other targets that they might mention) cannot be examined. Also, there is no discussion of which sWGA method they compare to - I believe there are a range of primer sets? I think more clarity on the different methods will be helpful for other researchers.

3. The authors comment on potential loss of infection complexity owing to sample processing. I appreciate the small sample size and hence ability only to report on trends but is this something that could be investigated more directly and presented? For example, were the two polyclonal infections both leukodepleted or both from hybrid capture? This adds to useful information for other researchers planning similar studies as per 2.

4. On the note of infection complexity, I wondered to what extent the REALMCCOIL analysis might be affected by the small baseline population despite the use of simulations? Is there any information on simple measures such as numbers/proportions of het positions?

5. Staying on the note of COI, it would be interesting to see the average read depth for the Pf samples versus the Po samples to understand if this might be a factor.

6. Regarding the population-level diversity, I wondered how the estimates might be affected by

geographic bias and if this could be explored further. For example, where there are multiple samples from a single study (e.g. 5 Pow from one study in Ethiopia, 4 Pow from one study in DRC and 5 Poc from another study in DRC etc) what would happen if only one representative were taken from each study?

7. There was no description on the apicoplast and mitochondrial markers - can the authors comment briefly?

Response to Reviewers' Comments

Key comments are underlined, and the authors' responses and changes to manuscript text are beneath in bold.

Reviewer #1 (Remarks to the Author):

Major Points:

1. I am concerned that the conclusions are drawn based on relatively few genomes (16 Pow and 13 Poc) to be highly informative of sub-Saharan Africa. Only African samples have been considered, and both Pow and Poc are present in other regions of the world. They would need to be included in the analysis to make claims of “comprehensiveness” and for robust interpretation. Indeed, the authors allude to this limitation about the sample size and the study limitations (Line 368). I am wondering why a few other publicly available WGS datasets on both Pow and Poc have not been included in the study that were generated as part of the first two studies describing the genomes of Pow and Poc (PMID: 27392654; PMID: 28117441) in 2016-17 and more recently (2024) in PMID: 38360879. Moreover, I would have liked to see parasite samples from a wider region in the world included in such a study, including parasites from the Indian subcontinent, SE Asia, and other parts of Africa. While establishing access to such globally representative samples may be challenging and time-consuming, I would have liked to see more samples, at least from the African continent, included in the study to make the observations robust and “comprehensive”.

Response:

Genomic data from the cited studies have now been included in these analyses, including new coverage of South Sudan, Nigeria, Congo, Sierra Leone, Kenya, Gabon, and Ghana. This brings the total number of African isolates up to 45 (21 *P. ovale curtisi*, 24 *P. ovale wallikeri*), excluding five isolates from Higgins et al. 2024 (PMID: 38360879) that were not incorporated due to low genome-wide coverage. Unfortunately, we were unable to obtain Asian *P. ovale* isolates for sequencing. The methods section, results, and all figures and statistics have been updated to reflect the new sample set.

2. Perhaps one of the most interesting things is the clear signal of the selective sweep on the dhfr-ts locus; however, that has been reported twice before. In this regard, the authors are missing a major reference (PMID: 38761813), who also performed a population genetics analysis of dhfr mutations in 518 *P. ovale* spp samples (314 Poc and 204 Pow) from Benin, Gabon, and Kenya and found selective sweeps in the dhfr locus (as did Chen et al, who they do reference).

Response:

The article cited as a major missing reference was published following the original submission of our original manuscript. We agree it is important to cite

this publication now that it has come out and have updated our background, results, and discussion accordingly. The background now reads:

(Background, Paragraph 3)

Original text: signs of a selective sweep involving a mutant *dhfr* allele (implicated in pyrimethamine resistance) were detected in *Poc* cases imported to China from Africa, though not in *Pow* [Chen et al.]

New text: signs of a selective sweep involving a mutant *dhfr* allele (implicated in pyrimethamine resistance) have been detected in both *Poc* and *Pow* [Chen et al., Joste et al.]

We have also incorporated a new supplemental figure 3 showing how three *Pow ts-dhfr* haplotypes described in Joste et al. are distributed across continental Africa. Compared to Joste et al., we describe a higher prevalence of non-wild type isolates, including the putative pyrimethamine resistance haplotype (Phe57Leu + Ser58Arg) at 37% frequency (7/19) in East and Central Africa (compared to 29% in Joste et al.).

3. I am also wondering if the *Pow* and *Poc* genomes (*Poc*221 and *Pow* 222) described in the Higgins et al. study (reference 15) could be used as a reference to capture better coverage and, hence, SNP statistics.

Response:

We agree it would be ideal to be able to use the novel reference genomes published earlier this year by Higgins et al. However, the hybrid capture we used to enrich genomic DNA in 21 samples we sequenced was designed to amplify regions in the nuclear chromosomes of the *Pow*CR01 and *Poc*GH01 reference genomes. Poor coverage of the newly-assembled regions in the novel references would lead to underestimated SNP density, so we are necessarily limited in our analyses to the chromosome assemblies which would be covered by our enrichment method.

4. In the discussion (line 386), the authors claimed that they 'shed light on the evolutionary history of the two *Plasmodium ovale* species'. This is not entirely true. What 'evolutionary history' have they uncovered here with 29 isolates representing two sympatric species? This is somewhat hyperbolic, as this study does not do this.

Response:

We agree that our original phrasing overstates our findings. We have softened our language in this section. It now reads:

(Discussion, final paragraph)

Original text: This analysis sheds new light on the evolutionary history of the two *Plasmodium ovale* species sympatrically circulating in sub-Saharan Africa while

indicating selective pressures on genes related to sexual differentiation, drug response, and immune evasion.

New text: This analysis provides a comparative genomic analysis of the two *Plasmodium ovale* species sympatrically circulating in sub-Saharan Africa and presents new evidence of selective pressures on genes related to drug response, sexual differentiation, and immune evasion.

5. Nomenclature: I would like to draw attention to the recently revised nomenclature of these two sympatric species as *Plasmodium ovalewallikeri* and *P. ovalecurtisi* (PMID: 38040603) and the subsequent debate associated with it (PMID: 38160179; PMID: 38272740). The authors mention that this nomenclature is 'currently evolving' (Line 82) but do not refer to these recent publications. I prefer the revised binomial nomenclature proposed by Snounou et al. (PMID: 38040603). However, I leave it to the editor to provide appropriate guidance.

Response:

We originally excluded these references due to the reference limit of 70, and the lack of consensus around naming. We have now added these references back in the revised manuscript (now references 16-18). We have opted to use the original names for consistency across the literature, and due to the aforementioned lack of consensus. However we tend to agree with Slapeta et al (<https://pubmed.ncbi.nlm.nih.gov/38160179/>) that it is necessary under ICZN rules to retain the binomial *Plasmodium ovale* for one of the two species.

Minor Points:

1. In line 140, the authors refer to Chromosome 10 of Pow as being underrepresented. Is it still the case that Chromosome 10 of Pow continues to be underrepresented despite all the genome sequencing and assembly efforts using a combination of various 4th-generation long-read sequencing technologies?

Response:

The novel references from Higgins et al. 2024 were able to construct a new *Pow* chromosome 10 with similar size to the *Poc* chromosome 10. However, as mentioned previously, the hybrid capture enrichment method we used for DNA enrichment prior to sequencing was designed using the original incomplete *PowCR01* chromosome, leading to the aforementioned lack of coverage across *Poc* chromosome 10. Thus, using the new reference genomes cannot overcome the lack of chr. 10 amplification due to hybrid bait design and would lead to similar poor coverage of *Poc* chr. 10 as well as the newly-assembled regions of *Pow* chromosome 10. We have clarified this in the results as below:

(Results, Paragraph 2)

Original text: The hybrid capture was originally designed for *P. ovale wallikeri*, with additional *P. ovale curtisi* baits then selected to cover areas that differ between the two *ovale* genomes. Due to the smaller sized *Pow* chromosome 10

(470kb), this approach did not provide coverage for the full *Poc* chromosome 10 (1,300kb).

New text: The hybrid capture was originally designed for *P. ovale wallikeri*, with additional *P. ovale curtisi* baits then selected to cover areas that differ between the two ovale genome assemblies (PowCR01 and PocGH01). Due to the *Pow* chromosome 10 being incomplete in the PowCR01 reference genome (only 470kb), this approach did not provide coverage for the full *Poc* chromosome 10 (1,300kb).

(Discussion, Paragraph 6)

Original text: Finally, *Poc* chromosome 10 was excluded from analyses due to poor coverage of hybrid capture baits.

New text: Finally, *Poc* chromosome 10 was excluded from analyses due to poor coverage of hybrid capture baits due to an incomplete PowCR01 reference genome used for bait design, and newly-assembled regions of *Pow* chromosome 10 now recognized in recent reference genomes similarly could not be examined.

2. Line 281: Figure 6C refers to Chromosome 14, yet the text gives Chromosome 13. Which is correct?

Response:

Thank you for catching this! Chromosome 13 is correct for Figure 6C. However, with the incorporation of new samples into the dataset, this particular region of the genome no longer shows the same strength of selective sweep. This portion of the figure has since been removed from the manuscript.

3. Line 297: Is this study really 'comprehensive'?

Response:

We have updated the text of the discussion as follows:

(Discussion, 1st paragraph)

Original text: We present the first comprehensive population genomics study of both *P. ovale* species from samples collected in sub-Saharan Africa.

New text: We present a comprehensive population genomics study of both *P. ovale* species from samples collected in sub-Saharan Africa.

4. Line 306: How confident are the authors of this? What evidence do they have for this claim?

Response:

Regarding line 306: "Since the time of our analysis, new reference genomes have been assembled from South Sudan and Nigeria that provide improved genome contiguity, annotation, and completeness,¹⁵ but are unlikely to change our main findings."

Higgins et al. note that the old and new reference nuclear chromosomes share 84.1% homology in *Poc* and 81% homology in *Pow*, with “most contiguity gains...in sub-telomeric regions” of the nuclear chromosomes, which were excluded from our analyses (see Methods). While these novel reference genomes may enable detection of signatures of selection at loci that were not originally assembled to the nuclear chromosomes, there is no evidence that any of these loci would change our conclusions. Indeed, we would only expect our main findings to change if the newly assembled regions differed substantially from the already assembled regions. As mentioned previously, because the hybrid capture and sWGA DNA enrichment methods were designed to amplify the original reference genomes, it is unlikely that these newly-assembled regions would show sufficient sequencing coverage and depth for robust analyses. Future studies might employ these novel reference genomes in the development of their enrichment methods to enable characterization of the newly-assembled regions.

5. Line 319: The timing of the divergence of these two species is questionable but is presented here as a fact.

Response:

Regarding line 319: “Relatively low nucleotide diversity in *Pow* may indicate reduced effective population size, increased inbreeding, or a population bottleneck in the time since *Poc* and *Pow* diverged 1.3 million years ago.”

Thank you for pointing this out. We have updated the text to better represent the uncertainty of divergence. It now reads:

(Discussion, middle of 2nd paragraph)

Original text: Relatively low nucleotide diversity in *Pow* may indicate reduced effective population size, increased inbreeding, or a population bottleneck in the time since *Poc* and *Pow* diverged 1.3 million years ago.

New text: Relatively low nucleotide diversity in *Pow* may indicate reduced effective population size, increased inbreeding, or a population bottleneck in the time since *Poc* and *Pow* diverged between 1.3 and 20.3 million years ago.

6. Line 353 How, exactly, do the authors propose that “Functional evaluation of different 353 alleles to determine their capacity to confer drug resistance, as well as monitoring of these alleles across the parasite populations over time”, will be “necessary to ensure that *P. ovale* do not develop resistance to prophylactic malaria treatment.” *P. ovale* spp. are already developing resistance.

Response:

While detection of selective sweeps in *Poc ts-dhfr* by Chen *et al.* and, more recently, *Pow ts-dhfr* by Joste *et al.*, are suggestive of evolutionary pressures on these loci, resistance *in vivo* has not been confirmed for either species. However, we agree that these surveillance studies suggest that resistance may be developing in the field, and we have adjusted our language to avoid implying that development of resistance is not of concern:

(Discussion, Paragraph 4)

Original text: Functional evaluation of different alleles to determine their capacity to confer drug resistance, as well as monitoring of these alleles across the parasite populations over time, will be necessary to ensure that *P. ovale* do not develop resistance to prophylactic malaria treatment

New text: Functional evaluation of different alleles to determine their capacity to confer drug resistance, as well as monitoring of these alleles across the parasite populations over time, will further clarify how interventions targeting *P. falciparum* may be simultaneously rendering *P. ovale* parasites harder to control.

Reviewer #2 (Remarks to the Author):

1. One of the main limitations is in the small sample size (good for *ovale* but small nonetheless!) and poor representation of some geographic regions. The authors note that the new study (not available when they conducted their analysis) by Higgins *et al* adds new geographic representation. I expect they may already be working on this but if not, I would suggest that they integrate the data from Higgins to improve the comprehension of their study.

Response:

We have incorporated 16 new samples with strong genome-wide coverage from studies by Higgins *et al.*, Rutledge *et al.*, and Ansari *et al.* Genomic data from the cited studies now additionally provide geographic representation of South Sudan, Nigeria, Congo, Sierra Leone, Kenya, Gabon, and Ghana. This brings the total number of African isolates up to 45 (21 *P. ovale curtisi*, 24 *P. ovale wallikeri*), excluding five isolates from Higgins *et al.* 2024 (PMID: 38360879) that were not incorporated due to low genome-wide coverage. All figures, tables, statistics, and methods have been updated to reflect inclusion of these genomes.

2. The loss of chromosome 10 from the hybrid capture samples is a shame but can't be changed at this point. I was a little concerned that in certain places the text still seems to promote hybrid capture over sWGA despite this problem. I would recommend being clearer in the messaging to other researchers who might want to conduct *ovale* genomics on the problems with the hybrid capture. For example, the authors don't mention it, but key drug resistance targets such as *mdr1* (and perhaps other targets that they might mention) cannot be examined. Also, there is no discussion of which sWGA method they compare to - I believe there are a

range of primer sets? I think more clarity on the different methods will be helpful for other researchers.

We appreciate this suggestion.

We have added text to the methods and discussion to introduce the sWGA approach used in the samples from Joste et al and highlight its potential compared to the limitations of hybrid capture:

(Methods, Library preparation and sequencing)

New text: Samples from Joste et al. and Higgins et al. included in our analysis were enriched using selective whole genome amplification (sWGA), employing sets of 5-10 primers designed to preferentially amplify the PowCR01 and PocGH01 genomes over human background DNA. (Clarke et al. 2017)

(Discussion, Paragraph 6)

Original text: Finally, Poc chromosome 10 was excluded from analyses due to poor coverage of hybrid capture baits. We do not expect this exclusion to systematically bias estimation of nucleotide diversity nor complexity of infection, though it does prevent us from evaluating chr. 10 loci for genomic signatures of selection.

New text: Finally, Poc chromosome 10 was excluded from analyses due to poor coverage of hybrid capture baits. We do not expect this exclusion to systematically bias estimation of nucleotide diversity nor complexity of infection, though it does prevent us from evaluating excluded loci (including *mdr1*, *msh3*, and *msh8*) for genomic signatures of selection. The availability of selective whole genome amplification protocols now provides a less expensive approach to targeted DNA enrichment for *P. ovale* spp. that does not rely on the specific design of hybrid capture baits.

3. The authors comment on potential loss of infection complexity owing to sample processing. I appreciate the small sample size and hence ability only to report on trends but is this something that could be investigated more directly and presented? For example, were the two polyclonal infections both leukodepleted or both from hybrid capture? This adds to useful information for other researchers planning similar studies as per 2.

The two polyclonal infections (one in each species) were hybrid captured samples with high genome-wide coverage. This increased coverage compared to a few of the leukodepleted and sWGA samples may improve the ability to detect polyclonal infections relative to other, lower-quality samples, though the remaining 19 hybrid capture samples with equally high coverage were still determined to be

monoclonal. Of the newly included genomes, 16/16 were also determined to be monoclonal in our analyses, though, as far as we can tell/based on reporting, these isolates were not specifically selected for sequencing based on low complexity of infection. We have added a sentence of the results to clarify the characteristics of the two polyclonal isolates:

(Results, Paragraph 6)

New text: These samples underwent hybrid capture and had high genome-wide coverage.

4. On the note of infection complexity, I wondered to what extent the REALMCCOIL analysis might be affected by the small baseline population despite the use of simulations? Is there any information on simple measures such as numbers/proportions of het positions?

We thank the reviewer for this question, as it is a fair point. First, the added sample size in this revision should hopefully improve our confidence in these estimates. Additionally, the number of heterozygous positions per sample is concordant with the calls of polyclonality. In *Poc* and *Pow*, respectively, the polyclonal samples showed 2,058 and 2,499 heterozygous calls while the remaining monoclonal samples had no more than 448 and 921 heterozygous calls.

5. Staying on the note of COI, it would be interesting to see the average read depth for the *Pf* samples versus the *Po* samples to understand if this might be a factor.

Response:

Average read depths across nuclear chromosomes in the three species were 44.1, 95.7, and 147.5 for *Poc*, *Pow*, and *Pf*, respectively. We have added this information to the legend of Figure 2 for context. We have also added a sentence to the discussion to address this limitation as below:

(Discussion, Paragraph 6)

New text: Disparate average read depth between the three species (44.1, 95.7, and 147.5 for *Poc*, *Pow*, and *Pf*, respectively) may also have differentially impacted our ability to detect polyclonal infections, but the low complexity of infection found in both *P. ovale* species should be robust given the overall satisfactory sequencing depth.

6. Regarding the population-level diversity, I wondered how the estimates might be affected by geographic bias and if this could be explored further. For example, where there are multiple samples from a single study (e.g. 5 *Pow* from one study in Ethiopia, 4 *Pow* from one study in DRC and 5 *Poc* from another study in DRC etc) what would happen if only one representative were taken from each study?

Response:

The inclusion of an additional 16 isolates in this revision should help to mitigate geographic bias driven by multiple isolates from a small number of studies. The revised manuscript now contains 21 *Poc* and 24 *Pow* isolates from 13 countries spanning East, Central, and West Africa.

Additionally, the finding of reduced nucleotide diversity in *Pow* relative to *Poc* was detected in both the overall analysis (Figure 3A) and in the comparison of 11 geographically-matched pairs of *Poc* and *Pow* isolates (Figure 3B). In the latter comparison, 8 out of the 11 pairs featured two isolates with the same study, country, and enrichment method. In the remaining three discordant pairs, coverage of the *Pow* isolate was either the same or higher than for the *Poc* isolate, which would bias *Pow* diversity upwards in contrast to our findings.

For this revision, we conducted a comparative analysis of nucleotide diversity using one isolate of each species from each of the ten included studies. These samples were matched by country when possible, and some studies only included isolates of one species. In the final comparison, featuring 8 isolates from each species, *Pow* showed significantly lower nucleotide diversity than *Poc* ($p < 0.0001$). This finding, in addition to the findings included in the manuscript, indicate that the observed lower nucleotide diversity in *Pow* relative to *Poc* is likely robust to bias from geographic location and source study.

7. There was no description on the apicoplast and mitochondrial markers - can the authors comment briefly?

The hybrid capture was not designed to amplify loci in the apicoplast and mitochondrial contigs, leading to low coverage of these regions in 21 hybrid capture samples and preventing analysis of their diversity, structure, and selection. We have now added this as a limitation of the study in the discussion.

(Discussion, Paragraph 6)

New text: The hybrid capture approach also did not enable enrichment and analysis of loci in the mitochondrial and apicoplast genomes, which were excluded from analysis.

REVIEWERS' COMMENTS

Reviewer #1 (Remarks to the Author):

The authors have done a respectable job of answering my queries addressing/clarifying some of my main concerns and addressing all of my minor concerns. The manuscript has greatly improved in its revised form. However, one of my original concerns about the main limitation of the study (small sample size to be truly representative of Sub-Saharan Africa) still stands even with the slightly expanded number (21 *P. ovale curtisi*, 24 *P. ovale wallikeri*) – with a few locations represented by only 1-2 samples.

Reviewer #2 (Remarks to the Author):

One of my main concerns in the original manuscript was the relatively small sample size (16 *Pow* and 13 *Poc*), which raised concerns on the robustness of the findings. Although still relatively modest, it is reassuring to see that the main findings presented in the original manuscript have been maintained with the increased sample size of 24 *Pow*, 21 *Poc*.

I also had some concerns about the skewed geographic representation within Africa but these have largely been addressed by the new samples and improvements to the comparative analyses, which incorporated more considered sample matching.

The authors have largely addressed all of my other queries/concerns and ensured to clarify the key limitations in the discussion, presenting a more transparent and balanced representation of the findings.

Reviewer #2 (Remarks on code availability):

I briefly reviewed the code. The README file is suitably informative and all of the scripts needed for processing appear to be available. However, I did not directly install and run the code.

Response to Reviewers' Comments

Key comments are underlined, and the authors' responses and changes to manuscript text are beneath in bold.

Reviewer #1 (Remarks to the Author):

The authors have done a respectable job of answering my queries addressing/clarifying some of my main concerns and addressing all of my minor concerns. The manuscript has greatly improved in its revised form. However, one of my original concerns about the main limitation of the study (small sample size to be truly representative of Sub-Saharan Africa) still stands even with the slightly expanded number (21 *P. ovale curtisi*, 24 *P. ovale wallikeri*) – with a few locations represented by only 1-2 samples.

We appreciate the reviewers note of the limitations of the study in both sample size and geographic coverage, and these are described in the discussion section.

Reviewer #2 (Remarks to the Author):

One of my main concerns in the original manuscript was the relatively small sample size (16 Pow and 13 Poc), which raised concerns on the robustness of the findings. Although still relatively modest, it is reassuring to see that the main findings presented in the original manuscript have been maintained with the increased sample size of 24 Pow, 21 Poc.

I also had some concerns about the skewed geographic representation within Africa but these have largely been addressed by the new samples and improvements to the comparative analyses, which incorporated more considered sample matching.

The authors have largely addressed all of my other queries/concerns and ensured to clarify the key limitations in the discussion, presenting a more transparent and balanced representation of the findings.

No revisions needed.

I briefly reviewed the code. The README file is suitability informative and all of the scripts needed for processing appear to be available. However, I did not directly install and run the code.

No revisions needed.